# Spatial Patterns of Turbidity in Cartagena Bay, Colombia, Using Sentinel-2 Imagery

**Monica Eljaiek-Urzola** [1,*], **Lino Augusto Sander de Carvalho** [2], **Stella Patricia Betancur-Turizo** [3], **Edgar Quiñones-Bolaños** [1] **and Carlos Castrillón-Ortiz** [4]

1. Faculty of Engineering, Universidad de Cartagena, Cartagena 130015, Colombia; equinonesb@unicartagena.edu.co
2. Department of Meteorology, Federal University of Rio de Janeiro (UFRJ), Rio de Janeiro 21941-916, Brazil; lino.sander@igeo.ufrj.br
3. Centro de Investigaciones Oceanográficas e Hidrográficas del Caribe, Cartagena 130001, Colombia; sepabe77@gmail.com
4. Faculty of Engineering, Civil Engineering Program, Universidad de Cartagena, Cartagena 130015, Colombia; ccastrillono@unicartagena.edu.co
* Correspondence: meljaieku@unicartagena.edu.co

**Abstract:** The Cartagena Bay in Colombia has vital economic and environmental importance, playing a fundamental role in both the port and tourism sectors. Unfortunately, the water quality of the bay is undergoing a deterioration process due to the significant influx of sediment from the artificial channel known as Canal del Dique. Although field campaigns are carried out semiannually with 12 monitoring stations to evaluate these impacts, understanding the spatial dynamics of suspended solids in the bay remains a challenge. This article presents a spatial analysis of water turbidity in the Cartagena Bay during the years 2018 to 2022, using Sentinel-2 images. To achieve this objective, an empirical algorithm was developed through the Monte Carlo simulation. The validation of the algorithm demonstrated an R-squared value of 0.83, with an RMSE of 2.72 and a MAPE of 24.93%. The results showed the seasonal variability, with higher turbidity levels during the rainy season, reaching up to 35 FNU, and lower turbidities during the dry season, dropping to 1 FNU. Furthermore, these findings indicated that the southern area of the bay presents the most significant turbidity variations. This research enhances our understanding of the bay's turbidity dynamics and suggests an additional tool for its monitoring.

**Keywords:** algorithm; turbidity; sentinel; Monte Carlo simulation

## 1. Introduction

Cartagena Bay is one of the most important estuaries in the north of Colombia. It holds a port complex with a high portion of Colombia's port traffic. Seaport activities represented 25% of the total cargo mobilized in the country in 2022, and Cartagena's port is the largest in the Caribbean region [1], becoming one essential port area in the Colombian Caribbean. Cartagena is a coastal city in the Colombian Caribbean with a population of around 1 million people; tourism is one of its main economic activities and what it is known for [2]. Cartagena Bay plays a crucial role in the tourism industry, as it is part of all the infrastructure associated with the city's tourism development, including beaches, docks, islands, and cruise ships [2]. On the other hand, Cartagena Bay has a distinctive environmental value due to the convergence of several ecosystems, such as mangroves, estuaries, deltas, coastal lagoons, beaches, coral reefs, areas of soft seabed on the continental shelf, and seagrass meadows. These ecosystems collectively represent an important biological value, characterized by the richness and diversity of flora and fauna, making them especially sensitive elements [3].

The Cartagena Bay exchanges water with the Caribbean Sea through two straits: the Bocagrande and the Bocachica. It receives fluvial water from the Canal del Dique (113 km), an artificial bifurcation of the Magdalena River (1528 km) that discharges approximately 1.7 Mt of sediments annually into Cartagena Bay (INVEMAR et al., 2020). Sediment influx results in a "river plume", which varies in direction and extension, ruling sediment dynamics within the Bay. Sediment transport studies in Cartagena Bay show that, during the months of greatest rainfall (September–October–November) of the rainy season (April–December), sediment discharge from the Canal del Dique into Cartagena Bay reaches its peak, and the plume covers nearly the entire Cartagena Bay area [4–7]. The bay surface turbidity variability ranges from 1.5 to 30 NTU for all seasons of the year, and the canal's turbidity levels within the bay range from 80 to 450 NTU, with an average of 211 NTU [4]. The inflow from the Canal del Dique and atmospheric wind patterns are the primary mechanisms responsible for the bay's hydrodynamics and, consequently, for shaping the turbid plume patterns within the bay [5,7,8].

Given its critical ecological and economic importance and the challenges posed by pollution, Cartagena Bay has been classified as one of the areas of the highest monitoring interest in the country, reflecting the need for ongoing efforts to preserve and protect its valuable ecosystems and biodiversity [9]. Consequently, the Bay of Cartagena is included in the waterbodies monitored semiannually by INVEMAR as part of REDCAM (Water Quality Monitoring Information System), which is an interinstitutional program related to marine and coastal pollution and serves as a valuable tool for the management and decision-making of the Environment Institution. This program includes the monitoring of 12 stations where in situ measurements of temperature, salinity, dissolved oxygen, and pH are carried out, using portable equipment. However, there is still a significant gap in the understanding of the turbidity spatial–temporal patterns within the Bay. A comprehensive understanding of the Canal del Dique plume and turbidity dynamics would help not only in measuring environmental impacts on the aquatic biota but also in the operation of ports and shipping activities.

Medium-resolution remote sensing sensors, such as SEAWIFS, MODIS, and VIRRS, have been used to understand and monitor water biogeochemical parameters worldwide [10–20]. However, due to the significant spatial variability, mainly in coastal waters, higher spatial resolution remote sensing sensors such as the Landsat Family, i.e., Thematic Mapper (TM) and Enhanced Thematic Mapper Plus (ETM+), and particularly the Operational Land Image (OLI) temporal series have been recently used to track water quality changes [21–26]. Since 2015, multispectral instruments (MSIs) on board the Sentinel-2 satellites have significantly improved spatial and temporal capabilities to study the behavior of biogeochemical parameters in several bays, including San Francisco Bay [27], Kastela Bay [28], the Bengal Bay [29], and Zhanjiang Bay [30]. The high spatial resolution of Sentinel-2 data makes them a valuable complement to traditional sampling strategies.

For the analysis of suspended solids or turbidity through satellite images, several empirical algorithms have been developed which contemplate the reflectance of a band or several bands measured by remote sensors on board satellites [31–33]. Some global semi-analytical algorithms have also been developed [34–36]. The semi-analytical algorithm developed by Nechad et al. [35] is a generic one-band algorithm designed for coastal waters with turbidity between 0 and 100 FNU, which was calibrated and validated for MERIS. The Dogliotti et al. [36] algorithm proposes a semi-analytical model that uses the reflectance of the near-infrared band (859 nm) to estimate turbidity in waters dominated by sediments with moderate-to-high turbidity, and the 645 nm band is used for waters with medium-to-low turbidity. This multisensor algorithm was developed for coastal and estuarine waters with turbidity levels ranging from 1 to 1000 FNU.

This work aims to understand turbidity patterns in Cartagena Bay through the use of remote sensing imagery. The research includes radiometric measurements, representing a pioneering effort in the Colombian Caribbean. In addition to developing a local, well-parameterized empirical algorithm to retrieve turbidity based on in situ measurements, the

study also focuses on applying MSI-Sentinel-2 images to understand the seasonal patterns in Cartagena Bay.

## 2. Materials and Methods

### 2.1. Study Area

Cartagena Bay is located in the northwest of South America, in the Colombian Caribbean (10°16′–10°26′N–75°36′–75°30′W) (Figure 1). It is separated from the Caribbean Sea by Tierrabomba Island and has a surface area of 82 km² approximately, with average and maximum depths of 16 and 26 m, respectively [37].

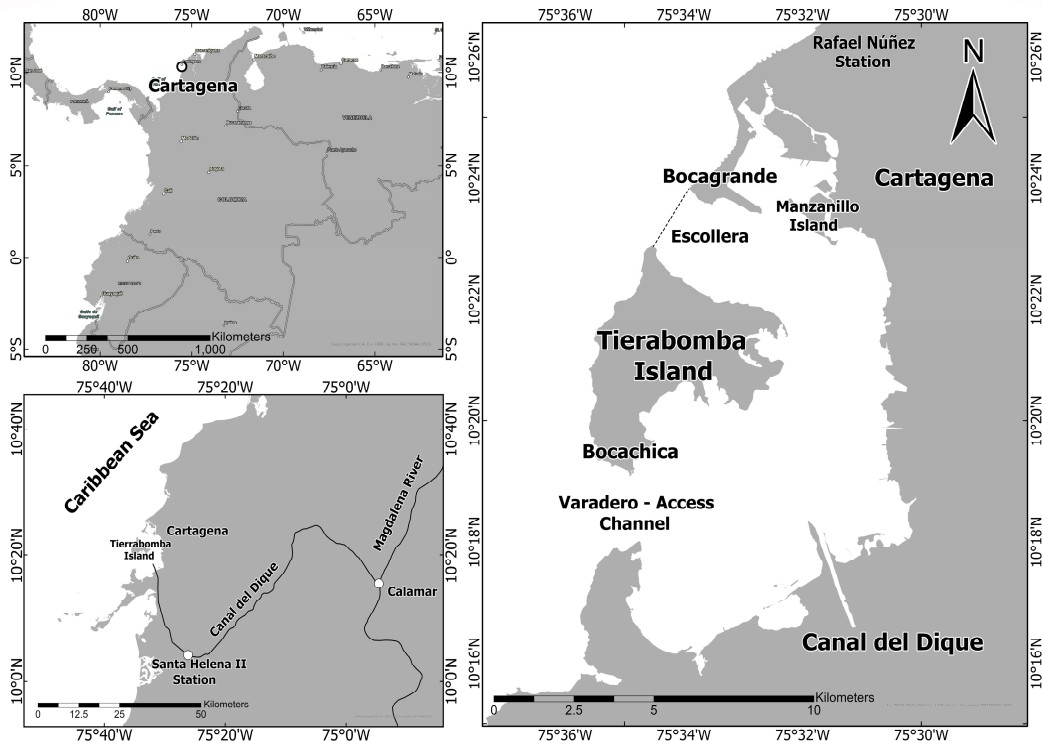

**Figure 1.** Cartagena Bay.

Cartagena bay has two straits: Bocagrande and Bocachica. The entrance to Bocagrande is partially obstructed by a submerged dike (Escollera), resulting in a depth between 0.6 m and 2.1 m. The Bocachica entrance comprises three narrow channels, with the deepest one, utilized as a navigation channel, measuring 100 m in width and 15 m in depth. The other two channels are shallower, with depths ranging from 0.5 m to 2 m and a width of 600 m each [38].

The main parameters that influence the hydrodynamics of the bay are the wind regime and the flow of the Canal del Dique. The multi-annual (2002–2022) average flow of Canal del Dique measured in Santa Helena II Station (Figure 1) is shown in Figure 2. Although this station is not located at the mouth of the canal, it provides valuable insights into the temporal fluctuations in the Canal del Dique's flows.

The climatological seasons in the Cartagena Bay area include a dry season from December to March and a rainy season from April to November, with the highest rainfall occurring from September to November [39] and the weakest winds with speeds below 2 m/s, while also seeing high flows from the Canal del Dique. During the dry season, characterized by prevailing NE trade winds, wind speeds peak at over 10 m/s, while low canal flows dominate. The months of March–April–May and June–July–August exhibit significant wind variability, with a notable intensification in July [3,5,40]. Cartagena's tide is classified as microtidal, mixed, and primarily diurnal [41].

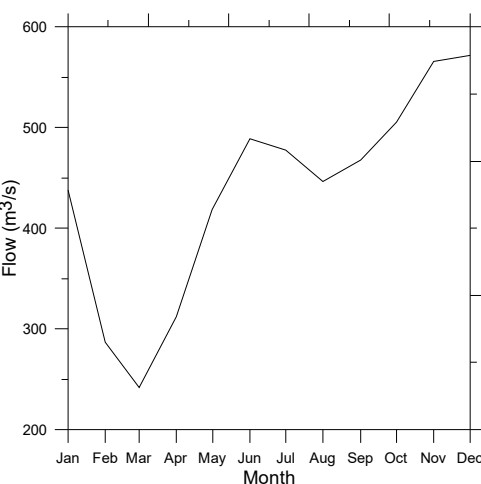

**Figure 2.** Multi-annual (2002–2022) average flow of Canal del Dique (Santa Helena II Station) [39].

The discharge rate of the Canal del Dique throughout the year significantly impacts the flux and reflux patterns of the current through the bay straits. During the season of high flows of the Canal del Dique, a predominant outflow of water masses occurs through Bocagrande, the shallow area of Bocachica, and the surface layer of the navigation channel [38]. Surface currents move from the northeast to the west in the central and northern bay, and frequent southwest winds sometimes generate intense south-to-north currents capable of transporting sediments to the north of the bay [7].

As discharge from the Canal del Dique decreases during the dry season, there is a change in exchange patterns. In Bocagrande, the weakening of the outward-directed flow and the possibility of reflux, especially during the rising tide, characterize this period. At the same time, in Bocachica, along the navigation channel, the currents directed toward the bay weaken as the tide falls [38]. Currents during this dry season are primarily influenced by the prevailing wind regime, with the tidal field contribution being insignificant due to the microtidal regime [7].

### 2.2. Field Measurements

The datasets were collected during two field campaigns: the first campaign, on 26 and 28 January 2022; and the second, on 2 and 3 February 2022. Both campaigns were carried out in the dry season due to favorable weather conditions and particularly low cloud cover. The sampling stations were distributed throughout the entire bay area, focusing on measurements within and outside the sediment plume. At all sampling stations, measurements of turbidity, Secchi depth, and radiometric data were concurrently collected, ensuring that these parameters were recorded simultaneously.

In the first campaign, data were taken from 13 stations, whereas, in the second campaign, data were obtained from 17 stations (Figure 3). In the first campaign, station selection was based on historical water quality data from the bay, ensuring coverage of the entire bay area, including the sediment plume. In the second campaign, four additional stations (ES1*, ES3*, ES4*, and ES5*) were strategically positioned within the sediment plume area to collect more data from this specific zone of the bay. The selection of these four additional stations was made in the field, taking into account the real-time location of the plume at the time of monitoring.

### 2.2.1. Turbidity and Secchi Depth measurements

Turbidity was measured in situ with a Hanna multiparameter probe (https://www.hannacolombia.com/productos/producto/hi-9829-medidor-multiparametro-impermeable-para-ph-ise-ce-od-turbidez-con-opcion-de-gps/, accessed on 5 November 2023) within the first 30 cm of the water column. Water transparency was measured using a Secchi disk

(white disk of 30 cm diameter). To ensure precise measurements from the boat, stability was a critical consideration when selecting the specific measurement station.

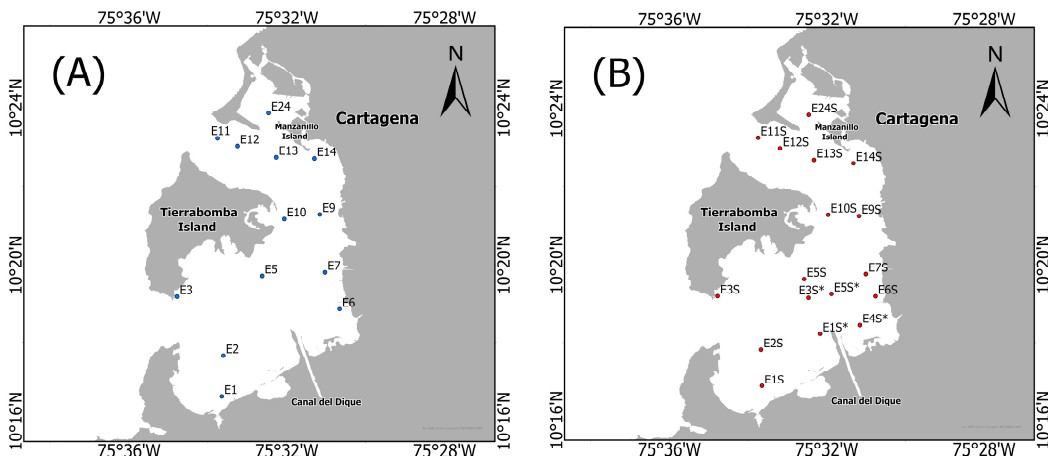

**Figure 3.** Spatial distribution of monitoring stations: (**A**) January 2022 and (**B**) February 2022.

2.2.2. Remote Sensing Reflectance

The above-surface method [42] was used for the estimation of remote sensing reflectance. This method establishes that the observation geometry must be at an azimuthal angle of $\Phi$ = 135 and a zenithal angle of $\theta$ = 40° and avoid the shadowing effect.

Radiometric measurements were taken at each station, using an Ocean HDX—Ocean Insight (https://www.oceaninsight.com/products/spectrometers/high-sensitivity/ocean-hdx/, accessed on 5 November 2023) spectrometer with a spectral range covering the spectrum from 400 nm to 1100 nm and according to Ocean Optics and Biogeochemistry Protocols for the validation of NASA satellite ocean color sensors—in situ optical radiometry [43]. Wind speeds were measured with a Davis Vantage Pro2™ portable meteorological station (https://www.estacionesdavis.es/davis-vantage-pro-2/10-davis-vantage-pro2-inalambrica.html, accessed on 12 December 2023) during radiometric measurements.

The estimation of remote sensing reflectance is calculated using Equation (1):

$$R_{rs}(\lambda) = \frac{(L_{water} - \rho * L_{sky})}{\left(\pi * L_{\frac{spectralon}{\sigma}}\right)} \tag{1}$$

where $L_{water}$ = the radiance measured pointing at water; $L_{sky}$ = the radiance measured pointing to the sky; $L_{spectralon}$ = the radiance measured pointing to the Spectralon; $\rho$ = the fraction of light reflected at the air–water interface with an angle of incidence of 40° and depending on the wind speed and the zenith angle; and $\sigma$ = the reflectance of a Lambertian reflector calibrated and supplied by the manufacturer (99% for the Spectralon used).

*2.3. Turbidity Algorithm*

The radiometric and water quality data acquired in the two field campaigns were merged into a single dataset. All the field data from the two campaigns were consolidated, resulting in two distinct groups of data: one for calibration and the other for validation purposes. Table 1 shows the stations used for calibration and validation of the algorithm. The data from station E12 were excluded due to uncertain measurements caused by shallow water depths.

**Table 1.** Distribution of stations.

| Campaign | Calibration Stations | | Validation Stations | |
|---|---|---|---|---|
| | Inside the Plume | Outside the Plume | Inside the Plume | Outside the Plume |
| January, 2022 | E7, E10 | E1, E2, E3, E6, E9, E14, E24 | E5 | E11, E13, E24 |
| February, 2022 | E1S*, E3S*, E5S* | E13S, E1S, E2S, E5S, E3S, E9S, E10S, E11S, E12S | E4S* | E6S, E7S, E14S |
| Total Stations | 5 | 16 | 2 | 6 |

Twenty-one in situ data pairs (turbidity and $R_{rs}$) were chosen for algorithm development: sixteen from areas outside the sediment plume and five from within the plume. Linear models based on $R_{rs}$ 665 and 865 were evaluated. The calibration of the turbidity empirical models was based on Monte Carlo (MC) simulations similar to the approach adopted by Augusto-Silva et al. [44] and Maciel et al. [45]. For this study, of the 21 samples designated for calibration, 10 were randomly selected and used for the calibration of the algorithm, considering a linear relationship between remote sensing reflectance ($R_{rs}$) and turbidity values (FNU). This process was repeated 300,000 times. The resulting values of the slope, intercept, and coefficient of determination ($R^2$) were recorded for each iteration.

For algorithm selection, the R histogram was utilized as a tool to identify equations, with this parameter in its most frequent range. Within this $R^2$ range, for each equation, the slope and intercept were determined, and the mode and standard deviations of the mode were calculated. Equations that had both the slope and intercept within a range based on the mode (mode $\pm$ standard deviation) were selected. From this group, equations with the best $R^2$ were chosen. The methodology applied for the calibration and validation of the empirical algorithm is shown in Figure 4.

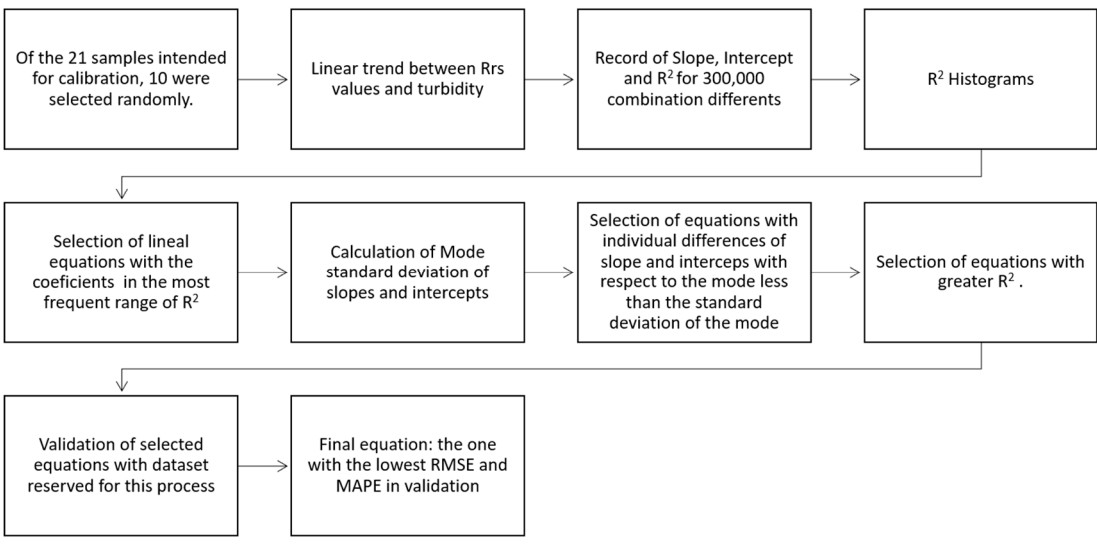

**Figure 4.** Methodology used for the calibration/validation of the turbidity empirical model.

For validation, the remaining eight samples were used. Calculated values from the selected models were compared with in situ values to assess the efficiency of the algorithms. The equation with the lowest estimated error was selected as the best.

Additionally, models presented in the Table 2 were tested in this study for algorithm intercomparison. The second column in Table 2 presents the algorithm formula and coefficients from the literature.

**Table 2.** Algorithms for intercomparison.

| Model | Algorithm |
|---|---|
| Dogliotti et al. [36] | $T = \frac{A_T^\lambda \rho_w(\lambda)}{\left(1 - \frac{\rho_w(\lambda)}{C^\lambda}\right)} \ [FNU]$ <br> $\lambda = 645 \ nm \ / \ 865 \ \text{nm}, \ \rho_w : \ \pi R_{rs}$ |
| Nechad et al. [35] | $T = \frac{A_T^\lambda \rho_w(\lambda)}{\left(1 - \frac{\rho_w(\lambda)}{C^\lambda}\right)} + B_T \ [FNU]$ <br> $\lambda = 665 \ nm \ A_T^\lambda = 282.95, \ C^\lambda = 0.1728 \ , \ B_T = 0.23$ |
| Kapalanga et al. [31] | $T = 15.31856 - 956.806(\rho_w 490) - 747.376 \ (\rho_w 560)$ <br> $+ 1742.455 \ (\rho_w 665) + 165.173(\rho_w 865)$ |
| Wang et al. [32] | $T = 8194.1 * x - 38.717$ <br> $x = (\rho_{w492} + \rho_{w865}) * \rho_{w865}$ |
| Cox et al. [33] | $T = 0.567 * x^{0.216}$ <br> $x = \rho_w 665 / \rho_w 490$ |

The results were compared with the in situ values to estimate the efficiency. To evaluate the performance of the algorithms, the root mean square error (RMSE) and the mean absolute percentage error (MAPE) were estimated with the following equations:

$$\text{RMSE} = \sqrt{\frac{1}{n}\sum_{i=1}^{n}(Tm_i - Tcal_i)^2} \qquad (2)$$

$$\text{MAPE} = \frac{1}{n}\sum_{i=1}^{n}\frac{|Tm_i - Tcal_i|}{Tm_i} \qquad (3)$$

where $T_{cal}$ and $T_m$ are the calculated and measured turbidity values for each sample *i*.

*2.4. Cartagena Bay Turbidity Spatial Analysis*

A spatial turbidity analysis was performed using 43 Sentinel-2A and -2B images filtered by cloudiness (one by month) and the turbidity algorithm selected. Images from October and November were excluded from the analysis due to their limited availability, making them less suitable for assessing turbidity during those months. The spatial variability of water turbidity was analyzed via the determination of the coefficient of variation. All the downloaded images were at level 1 format, and atmospheric correction (AC) was applied using Acolite software [46], which has been widely used for AC in turbid inorganic-rich environments [45,47,48].

The coefficient of variation quantifies the standard deviation as a percentage of the arithmetic mean (Equation (4)). To determine the coefficient of variation, a water mask of the bay was generated to calculate turbidity for each pixel in every image employed, using the developed algorithm. Subsequently, a grid of 500 m × 500 m was generated, totaling 332 points. Exclusions were made for grid points located over cloudy areas or in the shadow of clouds on the water's surface. Once the turbidity values were obtained for each point, organized by coordinates for each image utilized, the coefficient of variation was then computed.

$$C_v = \frac{\sigma}{\bar{x}} \qquad (4)$$

where $C_v$ = coefficient of variation, $\sigma$ = Standard deviation, and $\bar{x}$ = mean.

**3. Results**

Throughout the two field campaigns, north and northeast winds were predominant. The flow rates and wind speeds are presented in Table 3, reflecting typical conditions of the dry season.

**Table 3.** Canal del Dique flow rates and wind speeds during fieldwork.

| Date | Flow Measured at Santa Helena Station (m³/s) | Wind Speed Range (9 a.m.–12:30 p.m.), Rafael Nuñez Station (m/s) | Wind Speed Range (9 a.m.–12:30 p.m.) Measured during Fieldwork (m/s) |
| --- | --- | --- | --- |
| 26 January 2022 | 537.06 | NA | 1.8–5.0 |
| 28 January 2022 | 511.82 | NA | 0.5–4.0 |
| 2 February 2022 | 459.30 | NA | 1.3–5.0 |
| 3 February 2022 | 457.75 | 1.5-3.9 | 0.4–5.0 |

NA: not available.

### 3.1. In Situ Turbidity and Secchi Depth

The water turbidity ranged from 1.2 FNU to 31.3 FNU, while the Secchi depth varied between 0.2 m and 2.8 m. Most of the stations located inside the plume (Table 2) showed high turbidity levels (>12.1 FNU) and low Secchi depth values (<0.5 m), while most of the stations located outside the plume (Table 2) showed low turbidity values (<12.1 FNU) and Secchi depth values greater than 0.5 m, indicating a power-law relationship ($y = 4.3184 * -X^{-0.791}$), with $R^2 = 0.6735$), which can be observed when these variables are plotted and the strength and direction of the association (rs = $-0.828$) between turbidity and Secchi depth are measured (Figure 5). Similar results of this power-law relationship were observed in rivers, lakes, and reservoirs [49–51], whose behavior is very similar to that of the area influenced by the Canal del Dique. The relationship derived from the analysis of Secchi depth and turbidity aligns with the established scientific understanding, as the Secchi depth typically exhibits an inverse correlation with the concentration of total suspended solids (TSS) in aquatic systems [52].

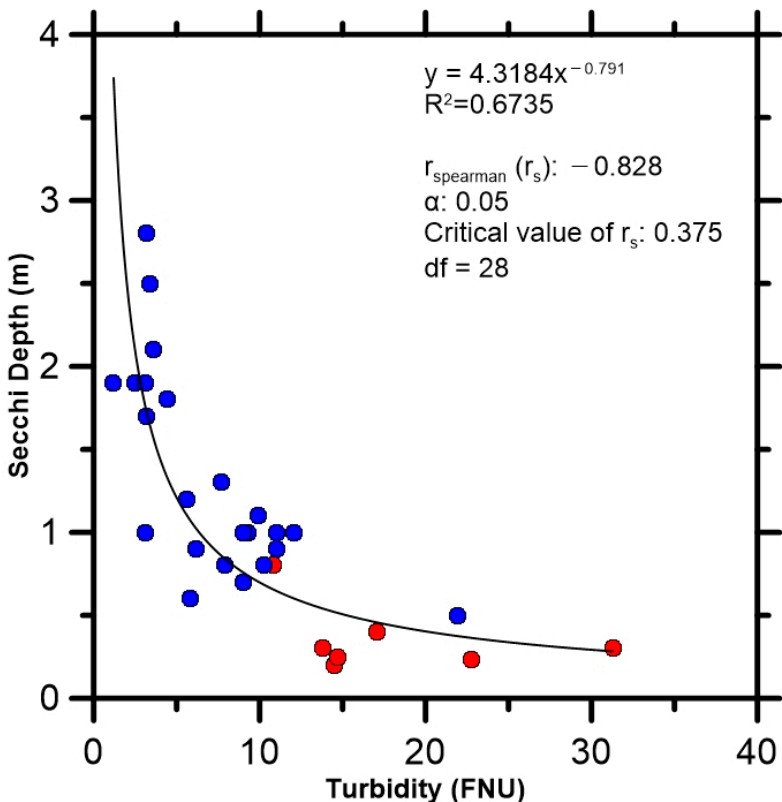

**Figure 5.** Relationship between turbidity (FNU) and Secchi depth (m). The red circles represent stations inside the plume, and the blue circles represent stations outside the plume.

### 3.2. In Situ Remote Sensing Reflectance

Two main groups of spectra associated with two areas of the bay could be observed from remote sensing reflectance calculated by in situ radiometric measurements, as illustrated in Figure 6. A zone of low-to-moderate turbidity was located in the stations outside the sediment plume, characterized by spectra with maximum remote sensing reflectance values between 550 nm and 600 nm (green) of the electromagnetic spectrum and with a second smaller peak in the range between 650 nm and 700 nm (red). This spectrum indicates the presence of estuarine waters with low or medium levels of chlorophyll and low total suspended solids [53–55].

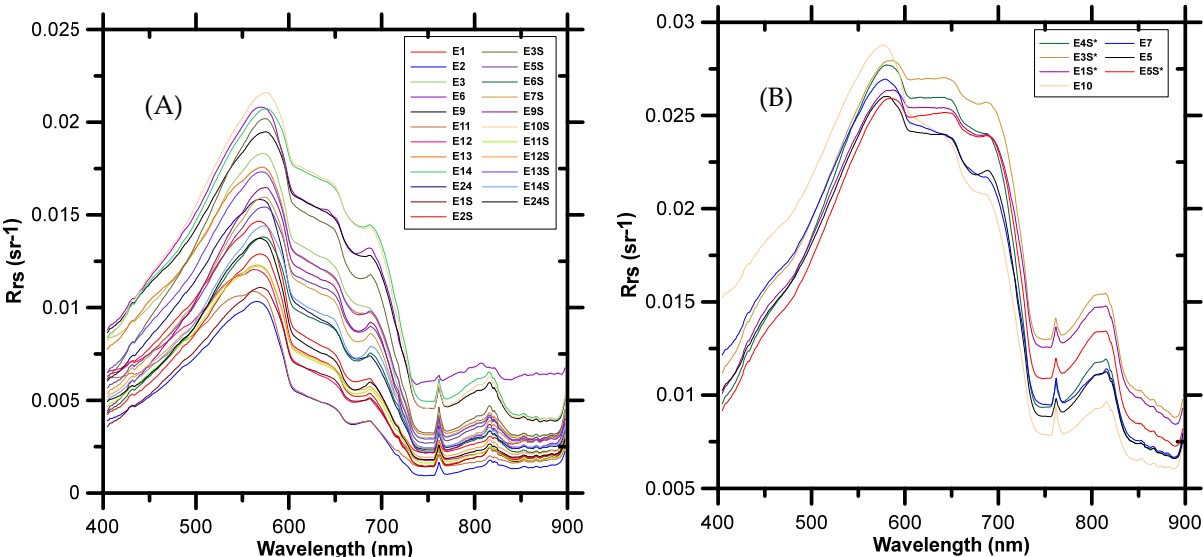

**Figure 6.** Remote sensing reflectance spectra (**A**) outside the sediment plume and (**B**) inside the sediment plume.

The second zone located within the sediment plume, characterized by moderate-to-high turbidity, shows a wider maximum extending from about 560 to 670 nm, with a steep slope of the spectrum between the blue and green wavelengths. This spectrum is characteristic of waters where the suspended particulate matter is dominated mainly by the mineral fraction and the organic fraction of non-living particles [53,55–57].

### 3.3. Turbidity Algorithm

#### 3.3.1. Calibration and Validation

The $R^2$ histograms from Monte Carlo simulation are shown in Figure 7. The most frequent $R^2$ range was between 0.65 and 0.7 for both of the models developed, with Rrs 665 and Rrs 865.

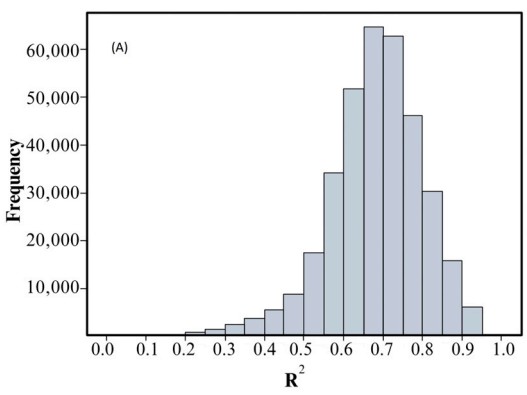
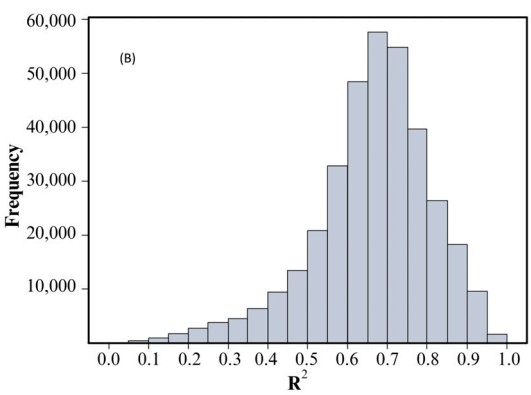

**Figure 7.** $R^2$ histograms from Monte Carlo simulation: (**A**) Rrs665 and (**B**) Rrs865.

The coefficients derived from Monte Carlo analysis of samples selected for calibration are shown in Table 4. Both models have a determination coefficient of $R^2 = 0.7$. Nevertheless, the model based on Rrs 665 was chosen due to its higher frequency of models in the range of 0.65–0.7 (>60,000) compared to Rrs 865 models (<60,000). Additionally, the model based on Rrs 665 presented lower values for both the RMSE and MAPE, reinforcing its superior performance. The selected algorithm is presented in Equation (5).

$$T = 707.98 \; * \; R_{rs}\,(665) + 0.03 \tag{5}$$

where $T$ is turbidity (FNU), and $R_{rs}$ (665) is the remote sensing reflectance at a wavelength of 665 nm. For validation, the calculated and the measured turbidity values were compared, obtaining an $R^2 = 0.834$, RMSE = 2.72, and a MAPE = 24.8 (Figure 8). It can also be observed that most of the data are within the confidence interval, indicating that the developed algorithm estimates turbidity values with a reliability close to 95%, which suggests that the selected algorithm works for Cartagena Bay.

**Table 4.** Coefficients derived from model calibrations.

| Wavelength (nm) | Intercept | Slope | $R^2$ | RMSE | MAPE |
|---|---|---|---|---|---|
| 665 | 0.03 | 707.98 | 0.7 | 3.1 | 25.9 |
| 865 | 0.77 | 2341.96 | 0.7 | 4.6 | 34.7 |

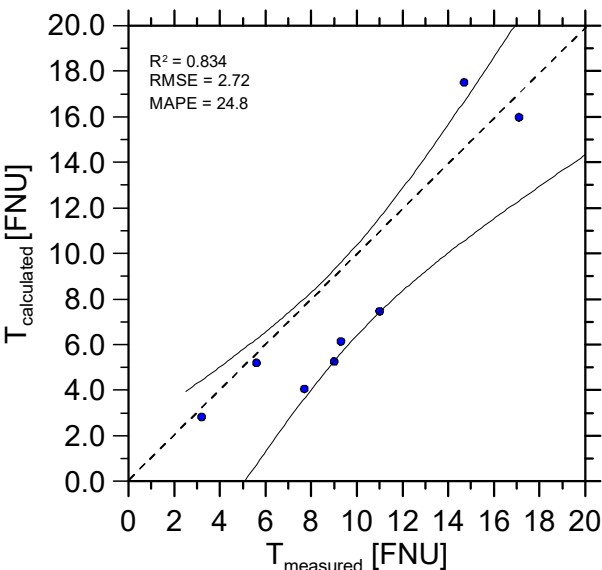

**Figure 8.** Scatterplot of algorithm-derived versus laboratory-measured (FNU) turbidity, corresponding to algorithm validation. The dashed line shows the 1:1 ratio, and the black solid line is the confidence interval (95%).

### 3.3.2. Comparison with other Algorithms

The performances of the algorithms were evaluated by applying them to in situ datasets, using their original algorithm coefficients. Statistical metrics are tabulated in Table 5. The evaluation of the proposed algorithms shows that the Cartagena Bay Algorithm has the best performance, with the highest $R^2$ (0.83) and the smallest RMSE (2.72 FNU) and MAPE (24.8%). The estimated turbidity from other algorithms from the literature shows higher uncertainties.

### 3.4. Turbidity Spatial Patterns

The average turbidity distribution, calculated for each month, from a series of five years, i.e., between 2018 and 2022, is shown in Figure 9. Regarding the spatial distribution in the bay of the turbid plume of the Canal del Dique, it is observed that the months of January and February show a homogeneous distribution of turbidity throughout the bay, with values below 5 FNU and with a reduced sediment plume with values of 18 FNU.

**Table 5.** Statistical analysis of turbidity values derived from the algorithms, using the datasets in this study.

| Algorithm | $R^2$ | RMSE | MAPE |
|---|---|---|---|
| This study: Cartagena Bay | 0.83 | 2.72 | 24.8 |
| Dogliotti et al. [36] | 0.75 | 31.4 | 107.9 |
| Nechad et al. [35] | 0.77 | 10.6 | 45.1 |
| Kapalanga et al. [31] | 0.72 | 15.6 | 188.3 |
| Wang et al. [32] | 0.80 | 43.2 | 565.2 |
| Cox et al. [33] | 0.79 | 83.3 | 93 |

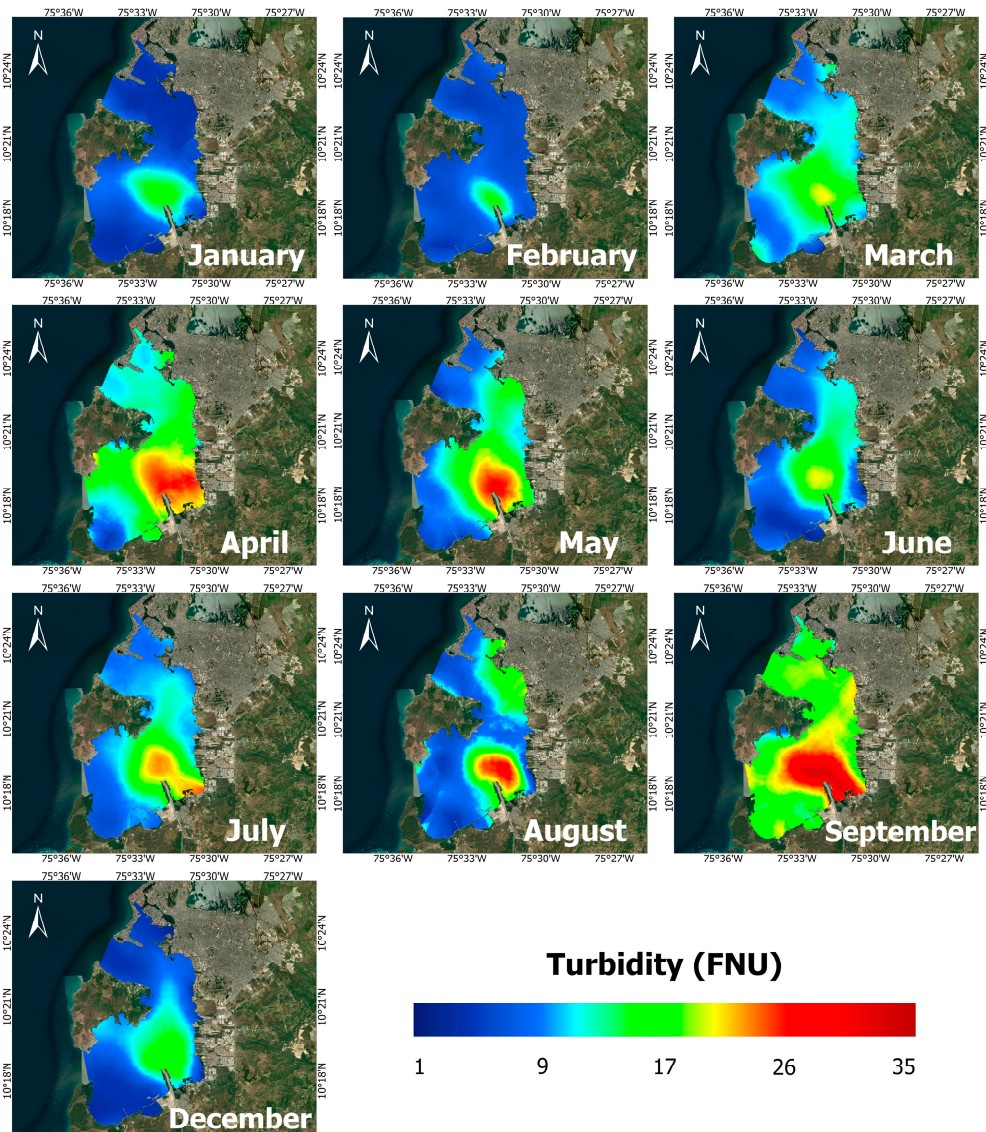

**Figure 9.** Water turbidity of Cartagena Bay, calculated from the empirical algorithm. The average of turbidity for each month in the time period 2018–2022.

In March, as the winds weaken, the onset of the rainy season becomes evident in the Canal del Dique basin. This transition is marked by a series of precipitation events, resulting in a notable increase in turbidity within the bay. Turbidity values during this period range between 3 FNU and 20 FNU. The southern area of the bay is predominantly affected by this increase in turbidity, characterized by the presence of filaments of turbid water near Manzanillo Island.

The spatial distribution for the month of April reveals significant inputs of turbid waters, with a plume oriented toward the southeast side, featuring turbidity values exceeding 25 FNU. Likewise, a bay highly disrupted by the influx of turbid waters is observed, extending to the northern area, with values ranging from 2 FNU to 18 FNU. A change in the turbid plume distribution pattern is observed in May, with maximum values (25 FNU) associated with the mouth of the Canal del Dique, which is reduced in extent compared to what was observed in April. In this month, a more stable gradient is evident, with a filament extending to the southern area of Manzanillo Island, featuring turbidity values of 17 FNU.

In June, the turbid plume of the Canal del Dique exhibits maximum values of 19 FNU, which are lower than those of the previous month, and the bay once again displays a homogeneous distribution toward the north and southwest areas of the bay, with turbidity values below 2 FNU. In July, a turbid plume similar to that of June is observed but with turbidity values exceeding 20 FNU. A very distinctive distribution pattern is observed in August, with two areas exhibiting maximum turbidity: one at the mouth of the canal, with peaks of 35 FNU, and another one toward the area near Manzanillo Island, with turbidity values ranging from 15 FNU to 18 FNU.

In September, the most significant disturbance of all the analyzed periods is observed, with a bay dominated by turbid water between 15 FNU and 35 FNU and a turbid plume concentrated on the eastern side and extending toward the northwestern side of the mouth. This dynamic marks the beginning of the heaviest precipitation in the basin [58] and, consequently, an increase in the flow, which intensifies in the months of October and November. Unfortunately, due to overcast conditions, it was not possible to create turbidity maps for these months. In December, the dry season begins, and, consequently, the plume distribution pattern decreases (<15 FNU), with turbidity values less than 2 FNU dominating the rest of the bay.

The seasonal variability of each year was analyzed according to the coefficients of variation in turbidity (Figure 10). The results show that the southwest and northeast areas are very dynamic, characterized by coefficients of variation ranging from 47% to 99%. On the contrary, the sediment plume area, represented in green tones, is less dynamic, with coefficients of variation between 25% and 40%.

The most significant variations were observed in 2018 and 2020, while the least variation occurred in 2022. This behavior in 2022 may have been influenced by the La Niña phenomenon, leading to a more consistent inflow of water from the Canal del Dique throughout the entire year [39]. During the dry season of 2022, a minimum flow in the Canal del Dique measured in Santa Helena Station II was observed around 350 m³/s [39], which exceeded the typical minimum flows in years under normal conditions, usually lower than 200 m³/s. Over the entire period from 2018 to 2022, the variation ranged between 30% and 78%. In general terms, the turbidity of the water presents a high level of variation, which indicates the dynamic and fluctuating character of the bay.

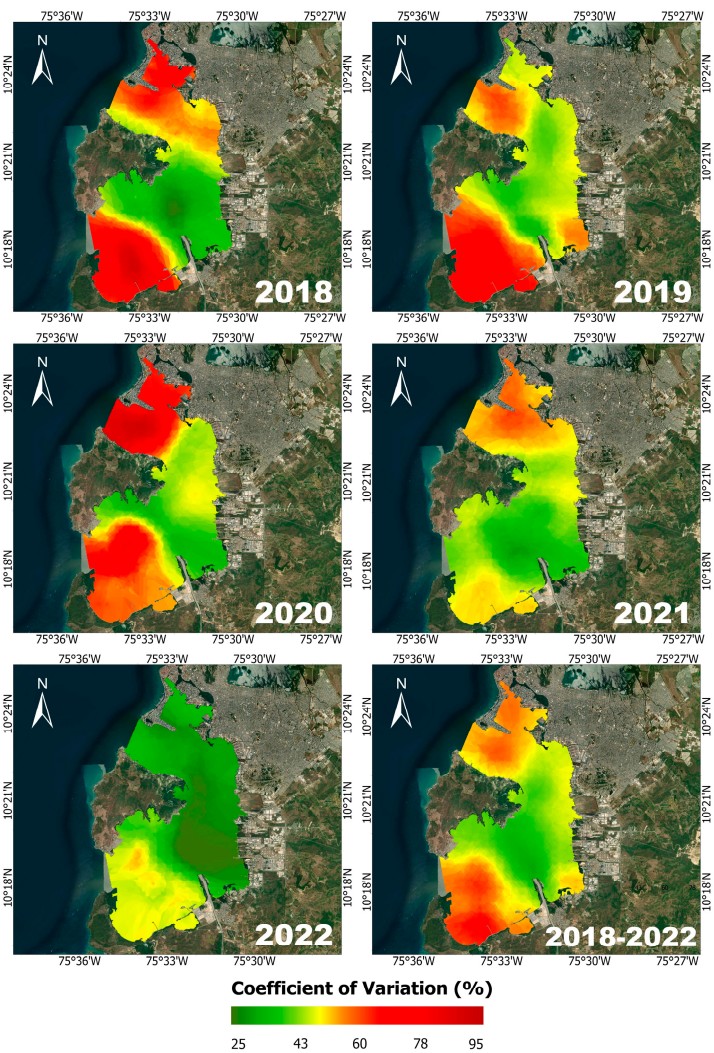

**Figure 10.** Variation in turbidity.

## 4. Discussion

### 4.1. Turbid Algorithm

The algorithm developed for Cartagena Bay demonstrated the strongest performance across the entire range of turbidities found in the bay (1–35 FNU). This result is expected, as empirical algorithms are often designed according to site-specific characteristics, requiring adjustment or adaptation of coefficients to achieve accuracy within a particular coastal region [43,56,57,59]. This algorithm uses a linear model that relates turbidity to remote sensing reflectance at a wavelength of 665. This approach is consistent with other research showing a strong correlation between the turbidity and reflectance of the bands located in the red area of the spectrum, particularly for turbidity values within the low-to-moderate range [43,58,60–62]. The lower performance of the near-infrared band algorithm is likely due to the low and moderate turbidity levels in the bay. This band is normally chosen for turbid waters, offering a reliable reflectance signal, since the 665 band can present saturation [60,63].

Based on the analysis of the algorithm performance and comparisons with others, it was observed that the Wang algorithm demonstrated a good performance, with an R$^2$ of 0.8. This performance could be attributed to the use of two different bands since algorithms based on several bands perform better in waters with high variation in suspended solid concentrations [32]. However, compared to the Cartagena Bay algorithm, it presents a greater error in the results, which may be due to the moderate variability in the water

turbidity and the optical constituents of the Bay of Cartagena. The Dogliotti algorithm had a good performance for waters with $\rho_w(645) < 0.05$, that is, when the reflectance of the red band is used for the model (low turbidity). For waters with $\rho_w(645) > 0.07$ (moderate-to-high turbidity), where the algorithm proposes the use of the NIR band or a blending scheme, the results were highly overestimated. The Nechad algorithm demonstrated a superior performance ($R^2 = 0.77$, RMSE = 10.6, and MAPE = 45.1) compared to the Dogliotti algorithm. However, the empirical algorithm developed in this research showed less uncertainty.

*4.2. Turbidity Patterns*

In general, the modeling with the algorithm reveals that the turbidity of the water in Cartagena Bay varies with the season. In the rainy season (April to November), the bay experiences higher turbidity levels, covering most of the bay area in September, with maximum values reaching up to 35 FNU. These turbidity fluctuations are primarily driven by the influx of sediment from the Canal del Dique. It is worth noting that these contributions peak during the rainy season, coinciding with the period when the Canal del Dique records its maximum flows (500–600 $m^3$/s during the rainy season), as measured at the Santa Helena II station [39]. The period of greatest rainfall coincides with the presence of weak winds and less intense currents in the region, making the entry of fresh water from the Canal del Dique an influential factor in the hydrodynamics of the bay. This results in a reduction in salinity and notable vertical density gradients, with flows from the mouth of the Canal del Dique directed toward the Bocachica and Bocagrande areas [8,38].

During the dry season (December to March), turbidity values typically fall within the range of 2 to 20 FNU, with higher values concentrated at the mouth of the Canal del Dique. This behavior is consistent with the hydrodynamic conditions of the season. The dry season coincides with increased wave activity, primarily driven by trade winds and the presence of cold fronts. During this period, contributions from the Canal del Dique decrease, and the vertical density gradient becomes less pronounced. These conditions facilitate the entry of more transparent oceanic masses of water through the two bay straits, aided by surface currents and tidal oscillations [7,8,38,58].

The results of this research support the conclusions of previous studies [4,5,7,64], which have also established that the wind regime and the flow of the Canal del Dique are the primary factors influencing seasonal variations in turbidity. As previously mentioned, the region with the most stable turbidity levels over time is the central area (sediment plume), where high turbidity levels persist consistently. In contrast, areas adjacent to the entrances of the bay present greater variability due to the fluctuations in the Canal del Dique inflow and the dynamics of water interchange through the straits [5,38].

**5. Conclusions**

This study demonstrates the efficacy of Sentinel-2 images for mapping and monitoring turbidity in Cartagena Bay. By employing the Monte Carlo simulation method, an accurate algorithm was established for estimating water turbidity, using field measurements. Around 300,000 combinations of turbidity and reflectance data pairs were evaluated for the bands studied (665 and 865), using linear regression models, which allowed us to obtain a robust model for determining turbidity. The results indicate that the algorithm based on the red band had a better performance for retrieving water turbidity, with a good validation accuracy of $R^2 = 0.83$.

The analysis of seasonal turbidity trends, carried out through Sentinel-2 images, showed the influence of seasonality: the increase in turbidity during the rainy season attributed to a greater discharge of sediments through the Canal del Dique. Additionally, the southwest and northeast regions of the bay showed the greatest variations, with coefficient-of-variation values ranging from 47% to 99%, while the sediment plume area showed the smallest fluctuations, with coefficient-of-variation values which ranged between 25% and 40%. This pattern may occur because the plume zone consistently maintains the highest

turbidity levels, while variability in other areas is predominantly influenced by the channel inflow and bay hydrodynamics, which vary seasonally.

The developed algorithm will serve as a valuable tool for the continuous monitoring of turbidity in Cartagena Bay and for the assessment of its water quality, including during upcoming sediment control initiatives. This research demonstrates the promising potential of using Sentinel-2 data to complement the understanding of turbidity dynamics in Cartagena Bay, offering a robust tool for continuous monitoring and evaluation of its water quality.

**Author Contributions:** Conceptualization, M.E.-U. and L.A.S.d.C.; funding acquisition, M.E.-U. and E.Q.-B.; methodology, M.E.-U. and L.A.S.d.C.; formal analysis, M.E.-U., S.P.B.-T., and L.A.S.d.C.; investigation, M.E.-U., L.A.S.d.C., and S.P.B.-T.; resources, M.E.-U. and S.P.B.-T.; writing—original draft preparation, M.E.-U.; writing—review and editing, M.E.-U., L.A.S.d.C., S.P.B.-T., and E.Q.-B.; visualization, C.C.-O. and M.E.-U. All authors have read and agreed to the published version of the manuscript.

**Funding:** This research and the APC were funded by the Research Vice President of the University of Cartagena, initiation and commitment record No. 130-2021—Resolution 00417 of 2021 and initiation and commitment record 084-2021—Resolution No. 00689 of 2021.

**Data Availability Statement:** Data are contained within the article.

**Acknowledgments:** The authors express their gratitude to the University of Cartagena and the Vice Rector for Research for providing financial support for the acquisition of equipment and fieldwork. In addition, the authors express their gratitude to the Hydrographic and Oceanographic Research Center (CIOH) and the students of the Environmental Modeling Research Group of the University of Cartagena for their invaluable support, which contributed significantly to the development of this project.

**Conflicts of Interest:** The authors declare no conflict of interest.

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
