# Peer review of "Spatial Patterns of Turbidity in Cartagena Bay, Colombia, Using Sentinel-2 Imagery"

_remotesensing, doi:10.3390/rs16010179_

Round 1

Reviewer 1 Report

Comments and Suggestions for Authors

The work is certainly a valuable compendium on how to develop an algorithm for assessing water turbidity with the support of remote sensing tools and data. However, the authors did not devote enough space to explaining channel inflow characteristics and bay hydrodynamics. Although the strong emphasis placed on the statistical explanation of the results obtained using various models was ultimately not confirmed by the physical conditions prevailing in the bay.

The introduction devotes a lot of space to a discussion of conditions in the bay rather than a solid review of the literature. The authors did not mention what wind conditions the samples were taken in. For the entire area preceding the sampling, it would be required to present wind conditions and currents moving water masses.

Therefore, it should be considered that the work is premature for further processing (rejected). The main criticism should be considered that the authors did not devote enough time to explaining channel inflow characteristics and bay hydrodynamics (currents, tides, water level, wind influence).

The presented results in the form of water turbidity patterns in the bay are not reliable.

General comments:

I have doubts whether this is the right goal. In particular, the sentence "This work aims to build a strategy to understand turbidity patterns in Cartagena Bay by the use of Remote Sensing Imagery" is very general and does not encourage the reader to further study this work.

Figure 1: The source of the map is not given unless the authors made it. Important details are missing from the map. Maybe some of them are unnecessary.

Figure 2: Quality is poor. It is worth integrating all the information about the location of objects in Figure 1.

Figure 3: Quality is poor. Repetitions in the title and drawing.

Figure 7: Y-axis data unreadable

Results

The results do not present what water flow conditions prevailed on the Canal or what wind conditions were recorded at the nearest meteorological station.

Discussion

It has not been sufficiently explained how the wind regime affects turbidity and the type of turbidity in a given season.

The discussion is a summary of the results and not a comparison with previous research by other researchers

Author Response

Dear Reviewer,

We have diligently addressed each comment and suggestion made by you, aiming to enhance the impact of our research. The detailed responses to the comments, along with the corresponding modifications made to the manuscript, are described in detail in the attached document.

We are confident that these revisions have strengthened the manuscript, aligning it more closely with Remote Sensing Journal standards.

Thank you once again for your comments and we hope that they have all been resolved in a good way.

Kind regards,

MONICA ELJAIEK-URZOLA

LINO AUGUSTO SANDER DE CARVALHO

STELLA BETANCUR-TURIZO

EDGAR QUIÑONES-BOLAÑOS

CARLOS CASTRILLÓN-ORTIZ

Reviewer 2 Report

Comments and Suggestions for Authors

This is a well written paper that is based on a significant body of work by the authors. The authors have obtained a tailored remote sensing algorithm tailored for the near shore waters of Cartagena bay and demonstrated its effectiveness. The comprehensive statistical analysis the authors use allows them to make a convincing case. The description of the ground truth field work is clear and complete and the results and their environmental implications are very well described and have significant potential impact. For these reasons I recommend publication of this work. However there are significant gaps and missing explanations in the description of the processes they used to arrive at their algorithm and these must be addressed in order to clear up potential confusions before publication can proceed.

The methodology shown in figure 5 must be discussed in much more detail in the text of the paper. A particularly troublesome point is that the authors mention Monte Carlo simulations without specifying what is simulated and by what model. Is this a remote sensing simulation using a full program such as Hydrolight or is this a direct Monte Carlo shuffling as required for a “Bootstrap” statistical analysis.  This comment applies even though the authors cite Silvia and Maciel in reference since at this time the paper does not make any sense as a self contained work without the obligation of reading these papers. At a minimum the substance of the analysis must be clearly described.

Line 188 “yi y xi” should be replaced by “yi and xi” since there are no “y” in either the RMSE or MAPE equations. Perhaps a notation using “Tci” for calculated turbidity at point i and “Tmi” for the measured turbidity at point I,  which would be a clearer notation.

“Rho w” used in all the formulas in table 2 which is presumably the normalized water leaving reflectance is not defined and it’s not mentioned that it is equal to “Pi Rrs” for a surface with diffuse/Lambertian properties. (Note the potential confusion for the reader with “Rho” defined on line 158 which is the water surface reflectance)

Line 140 What was the diameter of the Secchi disk. It should be specified

Some minor typos are noted below.

Line 20 “algorithm revealed” should be replaced bi:”algorithm demonstrated”

Line 31 “with high participation in” should be replaced by: “with a high portion of”

Line 65 “of Environment Institution” should be replaced by: “of the Environment Institution”

Line 69 “of turbidity” should be replaced by: “of the turbidity”

Line 121 “covertness” should be replaced by: “cover”

Author Response

Dear Reviewer.

We have diligently addressed each comment and suggestion made by you, aiming to enhance the impact of our research. The detailed responses to the comments, along with the corresponding modifications made to the manuscript, are described in detail in the attached document.

We are confident that these revisions have strengthened the manuscript, aligning it more closely with Remote Sensing Journal standards.

Thank you once again for your comments and we hope that they have all been resolved in a good way.

Kind regards,

MONICA ELJAIEK-URZOLA

LINO AUGUSTO SANDER DE CARVALHO

STELLA BETANCUR-TURIZO

EDGAR QUIÑONES-BOLAÑOS

CARLOS CASTRILLÓN-ORTIZ

Reviewer 3 Report

Comments and Suggestions for Authors

The authors proposed a new algorithm for calculating turbidity in the Cartagena Bay, which is an important ecosystem and economic resource in Colombia. The algorithm is based on analyzing data Sentinel-2, which has the advantages of high spatial resolution and a wide spectral range. The authors have developed a useful and reliable tool for water quality monitoring and demonstrated the applicability and effectiveness of the algorithm to investigate water turbidity in the Cartagena Bay. However, their study raises a number of questions that require further consideration.  

The authors used 20 calibration stations, of which only 5 were located inside the plume and the remaining 11 were outside it. For a 2-month observation period, this number of stations may be insufficient for a reliable evaluation of the algorithm's performance. Likewise, to test the algorithm, 8 stations were selected, of which 2 were inside the plume and 6 outside. It can also be observed that the spread of turbidity values at these stations was not large. As a result of the algorithm's operation, a standard deviation of 30% was obtained, which is a good indicator, but there is no guarantee that when the number of stations is increased, the deviation will remain the same.

The developed algorithm is stated to be capable of determining water turbidity in the range from 1 to 35 FNU, but the efficiency of the algorithm is demonstrated only for turbidities from 0 to 20 FNU, which is characteristic of the winter period in the study region. The authors also express their intention to apply the algorithm to continuous monitoring of water quality in the Bay, especially during times of increasing turbidity. In this case, it is necessary to calibrate and validate the algorithm not only with the data for the winter season, but also with data from other seasons when turbidity may reach higher values.  

In the paper, a comparison is made with other algorithms that have different applications and limitations. The Dogliotti algorithm was developed for turbidity in the range of 0 to 1000 FNU, but it is not recommended for use for low turbidity waters. In many studies, this algorithm shows poor agreement with in situ data in low turbidity waters. A more appropriate comparison would be with the Nechad algorithm developed for waters with turbidity between 0 and 100 FNU. The  Wang's algorithm was also created and calibrated for medium turbidity waters ranging from 0 to 600 FNU. The Abdelmalik algorithm, on the other hand, was calibrated for low turbidity lake waters, where the maximum turbidity was 5 FNU.  It is worth noting that of all the algorithms used for comparison, only the Dogliotti algorithm was calibrated on coastal waters, while the other algorithms were designed for lake waters. Perhaps because of these reasons, the values of the quality criteria for these algorithms turned out unrealistic.

Comments on the Quality of English Language

The English language needs minor editing

Author Response

(The authors gave the same response as above.)

Round 2

Reviewer 1 Report

Comments and Suggestions for Authors

The Authors improved the manuscript. I accept it in its present form. 

Reviewer 2 Report

Comments and Suggestions for Authors

The authors have answered all my questions to my satisfaction and the article can be published.